# Finetuning Weather Foundation Models to Develop Climate Model Parameterizations

## Abstract

Climate prediction models parameterize a range of atmospheric-oceanic processes like clouds, turbulence, and gravity waves. These *physical parameterizations* are a leading source of uncertainty and strongly influence future projections of global temperature rise. We present a fresh approach to developing parameterizations for coarse-climate models by leveraging pre-trained AI foundation models (FMs) for weather and climate. A pre-trained encoder and decoder from a 2.3 billion parameter FM (NASA and IBM's Prithvi WxC) — which contains a latent probabilistic representation of atmospheric evolution — is fine-tuned to create a data-driven predictor of atmospheric gravity waves (GWs). Current climate models are not fine enough to resolve GWs. We create an ML-based parameterization that learns GW fluxes from high-resolution "GW resolving" climate models to represent them in "GW missing" coarse-climate models. The fluxes predicted by our fine-tuned model are comprehensively evaluated using a set of three tests. Comparison with a baseline (Attention U-Net) reveals the superior predictive performance of the fine-tuned model throughout the atmosphere. The model outperforms the baseline even in regions excluded from the FM pre-training. This is quantified using the Hellinger distance which is 0.11 for the baseline and 0.06, i.e., roughly half, for the fine-tuned model. FMs are largely unexplored in climate science. Our findings emphasize their versatility and reusability to accomplish a range of weather- and climate-related downstream applications, especially in a low-data regime. These FMs can be further leveraged to create new parameterizations for other earth-system processes.

## 1 Introduction

Accurate prediction of future climate is a trillion-dollar challenge with critical consequences for the world economy, food security, global health, and urban planning. State-of-the-art future climate projections are highly uncertain. Obtaining reliable projections of future climate requires urgent improvements in existing climate models, many of which are strongly influenced by parametric uncertainty, scenario uncertainty, and structural uncertainty (Morrison & Lawrence, 2020; Lee et al., 2023). This study aims at demonstrating the untapped potential of AI foundation models to improve numerical climate models by addressing one of the leading sources of climate model uncertainty: physical parameterizations.

Foundation models (FMs) can be broadly defined as flexible task-agnostic models which are pre-trained using a self-supervised learning objective (Bommasani et al., 2022). Pre-trained FMs are then fine-tuned to perform a broad range of sub-tasks a.k.a. downstream tasks. A good example is OpenAI's ChatGPT, which is first pre-trained on large language datasets and is subsequently fine-tuned to perform several other language-related tasks.

FMs are largely unexplored in climate science. Only a couple of weather and geospatial FMs exist to date (AtmoRep (Lessig et al., 2023), ClimaX (Nguyen et al., 2023), and Prithvi (Jakubik et al., 2023)). Otherwise, the use of large AI models in meteorology, like FourCastNet, PanguWeather, and GraphCast (Pathak et al., 2022; Bi et al., 2023; Lam et al., 2023), is mostly restricted to weather prediction. Simply put, despite substantial training costs, these models have been limited to accomplishing just one task: medium-range weather forecasting. In this study, we use a state-of-the-art

FM, Prithvi WxC (Schmude et al., 2024), for weather and climate applications and apply it for the downstream task of developing data-driven physical parameterizations for climate models.

**Background:** Numerical climate models couple together multiple components of the earth system (atmosphere, ocean, land, ice, etc.) to predict climate evolution over years, decades, centuries, and beyond. Climate models often operate at a grid resolution of 100-300 km. This resolution is direly insufficient to even resolve the smaller-scale processes like clouds, precipitation, turbulence, gravity waves, etc. These processes are crucial for global energy balance. The traditional approach is to couple the computational fluid solver a.k.a. the dynamical core with a suite of *physical parameterizations* to crudely capture the unresolved effects (Alexander & Dunkerton, 1999; Lott & Miller, 1997; Bogenschutz et al., 2012; Iacono et al., 2000, to name a few parameterizations).

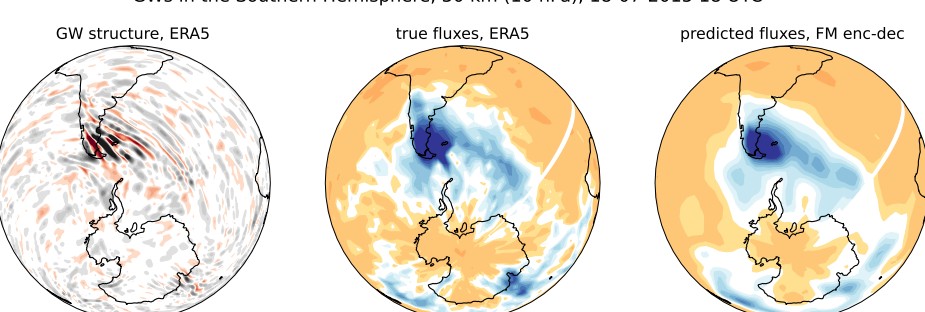

Figure 1: Comparing baseline model and the fine-tuned model. The left plot shows GWs over the Andes, parts of Antarctica and the Southern Ocean. The middle and right plots respectively show the true and predicted momentum flux ($u'\omega'$) carried by these waves. The vertical derivative of the flux equates to the net wind acceleration/deceleration applied by these waves.

As is discussed in Hourdin et al. (2017), most parameterizations are an inadequate representation of the respective process physics. They are subjected to a series of simplifications that compromise the process physics. This can be attributed partly to a limited process understanding and partly to the need to be efficient emulators that adhere to a prescribed climate model design. So, errors stemming from their design and parametric tuning often add up and result in inaccurate dynamics and momentum imbalances, leading to uncertainties in future climate projections (Golaz et al., 2013; Mauritsen et al., 2012; Zhao et al., 2018).

**Related research:** Using AI to learn from data and improve climate model parameterizations is an area of active research (see Mansfield et al., 2023; Eyring et al., 2024, for a review). Recent deep learning approaches include (a) learning process evolution from high-resolution models or parameterization data to represent it in coarse-resolution models (Espinosa et al., 2022; Chantry et al., 2021; Yuval & O'Gorman, 2023; Gupta et al., 2024b; Lu et al., 2024), (b) using equation discovery or similar techniques to learn analytical forms of sub-grid scale momentum closures (Zanna & Bolton, 2020; Jakhar et al., 2024), and (c) hybrid probabilistic combination of single-scenario high-fidelity data and multi-scenario low-fidelity data (Bhouri et al., 2023). Irrespective of the approach, the scarcity of high-resolution high-quality training data and low generalizability limits rapid progress.

**Contribution:** In this study, we introduce a fresh approach to promote the development of AI-driven climate model parameterizations. We blend the pre-trained encoders and decoders from the new Prithvi WxC foundation model with high-quality downstream task-specific data. We hence create a fine-tuned AI model that can skillfully predict and represent the missing subgrid-scale physics in coarse-climate models (which do not resolve the process). We demonstrate the effectiveness of our approach using atmospheric gravity waves (GWs) as a test process. The approach could be generalized to other processes like clouds and precipitation.

GWs are intermittent, small-scale (spatial scale $\mathcal{O}(1)$-$\mathcal{O}(1000)$ km) perturbations generated around thunderstorms, jet stream disturbances, strong flow over mountains, etc. (Fritts & Alexander, 2003).

GWs couple the different layers of the atmosphere by carrying near-surface momentum and energy to stratospheric and even mesospheric heights. GWs influence clear air turbulence, surface extremes, stratospheric circulation, and ocean heat transport. Thus, they are crucial to the earth's momentum budget yet are not resolved in climate models due to coarse grid resolution (Plougonven et al., 2020).

**Scientific importance:** a coarse $\mathcal{O}(100)$ km climate model practically misses all GW effects because it cannot resolve these small-scale waves (Achatz et al., 2023). So, we develop an ML model that learns GW effects *from a high-resolution climate model/data* (which resolves a substantial portion of the GW physics). This model can then be coupled to a coarse-resolution climate model to represent "missing" GW physics. This principle opens avenues to develop more physics-inclusive ML schemes to represent other missing processes (like clouds and precipitation) in coarse climate models.

**Instantaneous prediction**: our fine-tuned model skillfully predicts the GW momentum fluxes given the background atmospheric state, as shown in Figure 1. The structure of the excited GWs on 18 July 2015 is shown in Figure 1a. The model predicts the intermittent intensification of the fluxes around the Andes in South America and the Prince Charles Mountains in Antarctica. The intense fluxes over the Andes extend over to the Drake Passage and parts of the Southern Ocean, indicating that the finetuned model can learn and represent the lateral propagation and transient evolution of the generated waves; a physical feature absent in current model parameterizations (Kruse et al., 2022).

Our approach takes Prithvi WxC's high-dimensional latent space and clubs it with limited GW data from ERA5 to create a generalizable data-driven scheme for coarse-climate models:

- **Faster training, better performance, heterogenous data:** using pre-trained encoders and decoders allows faster training of the fine-tuned model compared to task-specific baselines. Despite different data sources for pre-training and finetuning, the fine-tuned model outperforms the baseline in predicting the global momentum flux distribution, regional flux distribution, and intermittent flux evolution.

- **Generalizes well to new regions:** the fine-tuned model outperforms the specialized baseline model even in the middle-to-upper stratosphere region where Prithvi WxC was not pre-trained.

- **Improved physics representation:** our scheme represents key aspects of GW physics which traditional climate model parameterizations do not: transient evolution (vs. steady-state evolution), and full three-dimensional evolution (vs. pure vertical evolution).

# 2 MODELS AND DATA DESCRIPTION

## 2.1 THE PRITHVI WXC FOUNDATION MODEL FOR WEATHER AND CLIMATE

Prithvi WxC, jointly developed by NASA and IBM, is a transformer-based deep learning architecture which combines ideas from several recent transformer architectures in order to effectively process regional and global dependencies of the input data and to efficiently process longer sequence lengths of tokens. This allows the model to, for instance, run in different spatial contexts or infuse additional tokens from off-grid measurements to the model during finetuning. Prithvi WxC has 2.3 billion trainable parameters and is trained on 160 atmospheric channels using 40 years of 3-hourly MERRA-2 reanalysis data at a $0.5° \times 0.625°$ spatial resolution.

The validation of Prithvi WxC extends from zero shot evaluations for reconstruction and forecasting to other downstream tasks such as downscaling of weather and climate models, the prediction of hurricane tracks, and climate model parameterization. The architecture of the pre-training backbone is shown in Figure 2. More details are provided in Schmude et al. (2024).

## 2.2 PREPARING TRAINING DATA FOR GW FLUX PREDICTION

The fine-tuning data for GW flux prediction was prepared using ERA5 global reanalysis data (Hersbach et al., 2020) retrieved at 25 km horizontal resolution, 137 vertical levels, and an hourly frequency. ERA5 resolves GWs with wavelengths longer than 150-200 km providing global, multi-decadal information on atmospheric GW evolution. Practically none of these waves are resolved by a typical climate model.

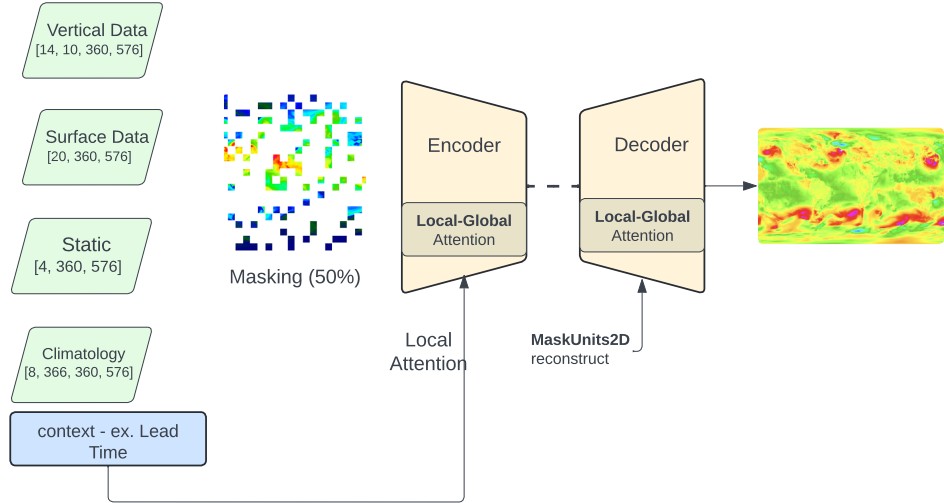

Figure 2: Pre-training model architecture for Prithvi WxC. The encoder and decoder blocks from Prithvi WxC are frozen and used for finetuning.

ERA5 does not provide GW momentum fluxes as output, they have to be computed. So, we compute the fluxes by applying Helmholtz decomposition (HD) (as in Lindborg, 2015; Köhler et al., 2023) on the raw ERA5 output as follows. First, the horizontal winds ($u$ and $v$) from ERA5 are decomposed into rotational and divergent components:

$$\vec{u} = (u, v) = -\nabla\phi + \nabla \times \psi \qquad (1)$$

where $\phi$ is the potential function such that $\nabla\phi$ is irrotational. Similarly, $\psi$ is the rotational stream-function function such that $\nabla \times \psi$ is non-divergent. $\phi$ and $\psi$ are used to reconstruct the divergent (div) and rotational (rot) parts of the horizontal flow as:

$$\vec{u} = (u, v) \xrightarrow{HD} (u_{div}, v_{div}) + (u_{rot}, v_{rot}). \qquad (2)$$

These are combined with the zonal mean removed vertical velocity ($\omega'$) to compute the directional GW momentum fluxes:

$$\vec{F} = (F_x, F_y) = g^{-1}(u'_{div}\omega', v'_{div}\omega'). \qquad (3)$$

which we aim to learn using the ML models. Here, $g = -9.81$ m/s$^2$ is the acceleration due to gravity.

The procedure is applied to create the finetuning training data. The top 15 out of the 137 vertical levels are discarded due to artificial model damping. All input-output pairs are coarse-grained from 25 km resolution to a 64 latitudes $\times$ 128 longitudes grid (roughly 280 km resolution) to obtain conservative wave averages. The fluxes are computed for four years: 2010, 2012, 2014, and 2015. This corresponds to roughly 35k training samples, which pretty much classifies as "data-scarce".

**Variables for baseline:** the input consists of winds $u$, $v$, potential temperature $\theta$, which is a function of temperature $T$ and pressure $p$ (in hPa) as $\theta = T(p/1000)^{-0.286}$, each on 122 vertical levels, 64 latitudes and 128 longitudes. Similarly, the output is fluxes $u'\omega'$ and $v'\omega'$, each on 122 vertical levels, 64 latitudes and 128 longitudes (Figure 3).

**Variables for finetuning:** slightly different from the baseline, the finetuning input consists of winds $u$, $v$, temperature $T$, and pressure $p$, each on 122 vertical levels, 64 latitudes and 128 longitudes. Similarly, the output is potential temperature $\theta$ (for validation), and fluxes $u'\omega'$ and $v'\omega'$, each on 122 vertical levels, 64 latitudes and 128 longitudes (Figure 4).

## 2.3 BASELINE MODEL

An advanced baseline compared to standard MLP was created by training an Attention U-Net model (Oktay et al., 2018) on the ERA5 data. The input is downsampled using four convolutional blocks and then upsampled using four convolutional blocks. The skip connection at each level comprises

learnable attention layers. For every downsample (upsample), the number of channels increases (decreases) by a factor of 2 but all spatial dimensions reduce (increase) by a factor of 2. As a result, the baseline model consists of over 35 million learnable parameters and provides a robust comparison benchmark for the finetuning model. The learning rate for the model was set to 0.0001. On a single A100 80 GB GPU the model took around 110 hours to complete 100 epochs.

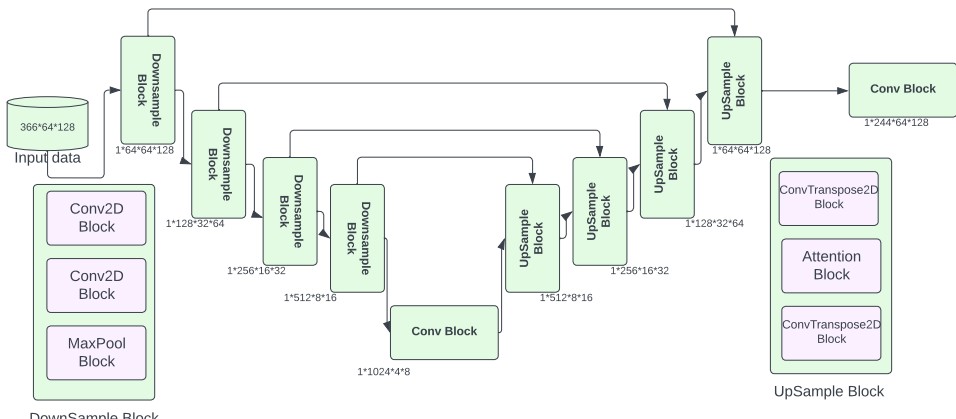

Figure 3: Model Architecture for the Attention U-Net baseline (schematically identical to Oktay et al. (2018)).

## 2.4 DESIGNING A FINETUNING MODEL

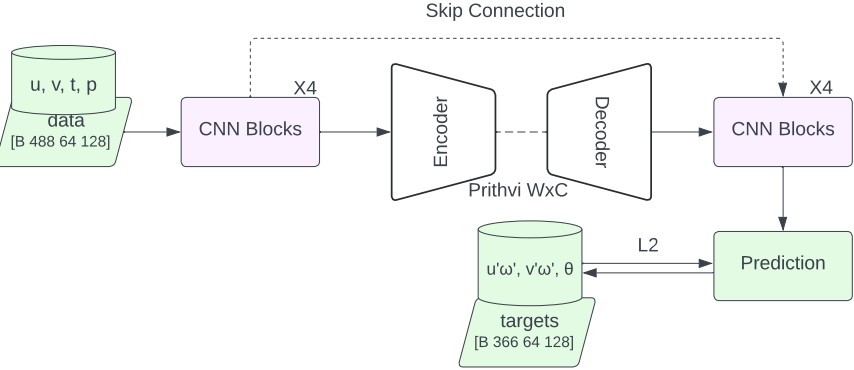

Figure 4: The finetuning architecture comprises of (in order) 4 learnable convolutional layers, the frozen encoder, the frozen decoder, and 4 more learnable convolutional layers. A skip connection connects the former and latter convolutional layers.

The architecture schematic for the finetuning is shown in Figure 4. During fine-tuning we freeze the encoder and decoder from Prithvi WxC. The frozen encoder is preceded by 4 learnable convolutional blocks each with an increasing number of hidden channels, i.e., $C$, $2C$, $4C$ and then $8C$, where $C$ = 160. Likewise, the frozen decoder is succeeded by 4 new learnable convolutional blocks. Since gravity wave flux prediction is an instantaneous flux calculation task, we fix Prithvi's lead time $\delta t$ to zero. The instantanous model input for fine-tuning has shape [1, 488, 64, 128] where the 488 channels comprise the four background variables $u$, $v$, $t$ and $p$ on 122 vertical levels each, and on a $64 \times 128$ horizontal grid, as discussed above. The model was fine-tuned to produce an output with shape [1, 366, 64, 128] comprising of the potential temperature $\theta$, and fluxes $u'\omega'$, and $v'\omega'$ on 122 vertical levels each. The model trained for 26 hours on 2 nodes of 4 A100 80GB each for 100

epochs, where each node had 4 A100 GPUs. However, the model error converged to lower than the baseline model error after just 40 epochs of training.

The model leveraged a U-Net like architecture to promote extracting high-frequency information from the source data. We re-emphasize that Prithvi WxC was pre-trained on the MERRA-2 dataset but the finetuning was accomplished using ERA5 data instead.

Both the baseline and the finetuned models use global information as input to predict global fluxes as output. This provides a strong contrast to traditional "single-column" climate model parameterizations. Access to the global atmospheric state allows the models to learn spatio-temporal correlations and horizontal propagation of gravity waves.

Both models were optimized using MSE Loss:

$$\mathcal{L}(\vec{x}, \vec{y}) = \frac{1}{n} \sum_{i=1}^{n} (x_i - y_i)^2 \tag{4}$$

## 3 RESULTS

We test both the steady-state distribution of the predicted fluxes and their evolution in time. The steady state distributions test how well our models generate the possible range of flux responses, which are crucial to modeling atmospheric extremes. Likewisetime evolutionlution tests their intermittent generatiotemporal coherenceerence. Here, we only show the results for the zonal flux $u'\omega'$. The fine-tuned model performs equally well for $v'\omega'$ while the baseline performance is worse. Equivalent plots for $v'\omega'$ are shared in the Appendix.

### 3.1 TEST 1: GLOBAL, STEADY STATE FLUX DISTRIBUTION

The observed and predicted global distribution of the GW momentum fluxes at different sampling frequencies is shown in Figure 5. The histogram represents the distribution of May 2015 monthly mean momentum flux over all the points in the troposphere and the stratosphere. Both the baseline and the fine-tuned models simulate the monthly mean distribution with remarkable accuracy both in the bulk of the distribution and its tails (Figure 5a). To quantify the difference between the two distributions, we use the Hellinger distance defined as follows.

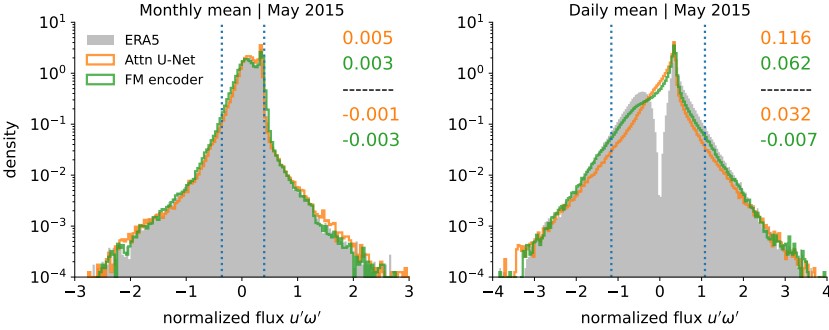

Figure 5: The distribution of the (left) May 2015 averaged and (right) daily averaged GW flux $u'\omega'$. Gray shading shows the true underlying distribution, orange the baseline prediction, and green the fine-tuning prediction. Numbers above and below the dashed line respectively indicate the Hellinger distance and tail-Hellinger distance for the corresponding predictive model. The dotted lines show the 2.5[th] and 97.5[th] percentile respectively.

**Hellinger Distance.** Given two probability densities, $p$ and $q$, their Hellinger distance, $\mathcal{H}$, is defined as:

$$\mathcal{H}(p, q) = 1 - \int_{x \in X} \sqrt{p(x)q(x)} dx. \tag{5}$$

By definition, $\mathcal{H} \in [0, 1]$. A Hellinger distance of 0 means the distributions are identical almost everywhere, while a Hellinger distance of 1 implies the distributions are disjoint, i.e., $p$ is non-zero

wherever $q$ is zero, and vice versa. Heuristically, we treat a Hellinger distance of 0.05 or less as pretty good.

**Tail-Hellinger Distance.** To specially quantify the accuracy around tails, we define an updated Hellinger distance for distribution tails, or the "tail-Hellinger" distance, between $p$ and $q$, $\mathcal{H}_{T,\epsilon}$ as:

$$\mathcal{H}_{T,\epsilon}(p,q) = \frac{1}{2} + \frac{1}{4\epsilon} \int_{x \in \mathcal{V}} p(x)dx - \frac{1}{2\epsilon} \int_{x \in \mathcal{V}} \sqrt{p(x)q(x)}dx. \tag{6}$$

Here $\mathcal{V} = (\infty, x_1] \cup [x_2, \infty)$ is a tail subset of $X$, and for cumulative distribution function $F$, $F(x_1) = \epsilon$ and $F(x_2) = 1 - \epsilon$. Unlike the regular Hellinger distance, the tail Hellinger distance can also be negative. A negative value would imply a fatter tail of the predicted distribution than the true distribution. For $\epsilon = 0.5$, the tail-Hellinger distance introduced here yields the regular Hellinger distance. More details are provided in the Appendix.

The baseline and the fine-tuned model have a Hellinger distance of 0.005 and 0.003 from the true distribution suggesting that the two distributions are nearly identical to the underlying ERA5 distribution. Albeit slightly negative, the tail-Hellinger distance too is almost negligible, indicating very similar tails.

To consider the time-evolving fluxes which may be averaged out in a monthly mean, we also considered the distribution of the daily sampled momentum fluxes (Figure 5b). The daily fluxes maintain high accuracy around the tails as the monthly mean (with slight degradation in the tail-Hellinger distance for the baseline (orange)), but the daily predicted fluxes from both models do not exhibit the minimum around 0 seen in the observed fluxes. Simply put, our models predict the bulk of the daily distribution quite well, but struggle a bit with predicting small flux values. This is reminiscent of prevailing problems with even the state-of-the-art weather prediction models which skillfully predict large-scale features but fail to project the same level of accuracy in predicting the small-scales. As a result of this deviation, for daily sampling, the baseline and fine-tuned models have a degraded Hellinger distance of 0.116 and 0.062 respectively. So, our fine-tuned model consistently outperforms the baseline model both on monthly mean and daily mean global statistics.

## 3.2 TEST 2: REGION-WISE, STEADY STATE FLUX SPECTRUM

The dynamical evolution of atmospheric GWs can notably vary with height, region (latitude and longitude), season, etc. The steady-state distributions conceal this. For a more stringent evaluation, we divide the global domain into 5 regions and 4 altitudes. The five regions comprise the two hemispheric poles, the two hemispheric midlatitudes, and the tropics. The four regions comprise the lower troposphere, the upper troposphere, the lower stratosphere, and the upper stratosphere.

The observed and predicted monthly mean distributions over the 20 slices are shown in Figure 6. The predicted and averaged fluxes agree quite well throughout the lower and upper troposphere. The Hellinger distances are consistently lower than 0.02 in most cases; an exception being the northern hemispheric poles in the upper troposphere. Within the troposphere, the baseline (in orange) has a slightly better Hellinger distance than the finetuned model but both models largely agree well on the captured distributions.

Compared to the troposphere, the Hellinger distances are higher throughout the stratosphere. The tropics and midlatitudes in the lower stratosphere have distances within our 0.05 threshold, but the polar regions have distances of up to 0.14. The baseline performance, however, gets much worse in the upper stratosphere with distances reaching up to 0.62 in the northern hemisphere upper stratosphere. In contrast, the distances for the fine-tuned model are constrained within 0.15. The fine-tuned model, thus outperforms the baseline in the stratosphere, with conspicuous differences.

The baseline model has a lower variance than the fine-tuned model even as the Prithvi WxC encoder-decoder model was not trained on upper stratospheric data at all (Figure 9). The performance improvement, then, can be attributed to the substantially higher volumes (20 years) of pre-training data as opposed to merely four years of ERA5 data for baseline. This allows for more mature development of the FMs latent space, leading to more effective learning during fine-tuning.

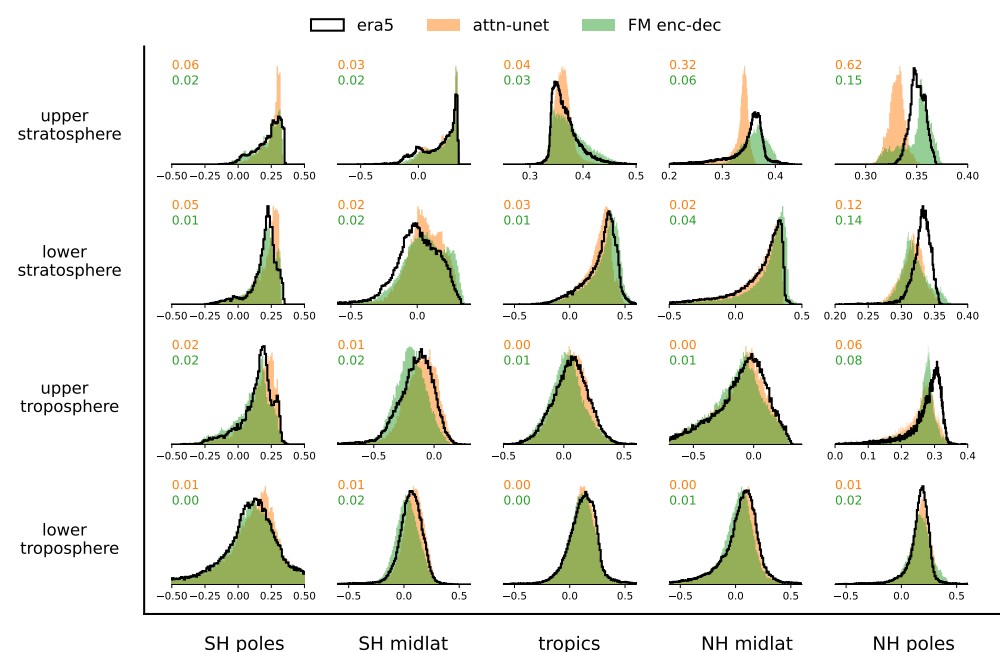

Figure 6: Global flux distributions segregated according latitude and height. The numbers indicate the respective Hellinger distances w.r.t the true distribution from ERA5 (black). For each latitude band, averaging is conducted over the whole latitude circle, i.e. over all longitudes.

## 3.3 TEST 3: INSTANTANEOUS, INTERMITTENT EVOLUTION OF GRAVITY WAVES

Our final, most stringent test assesses the time-evolving response of the models.

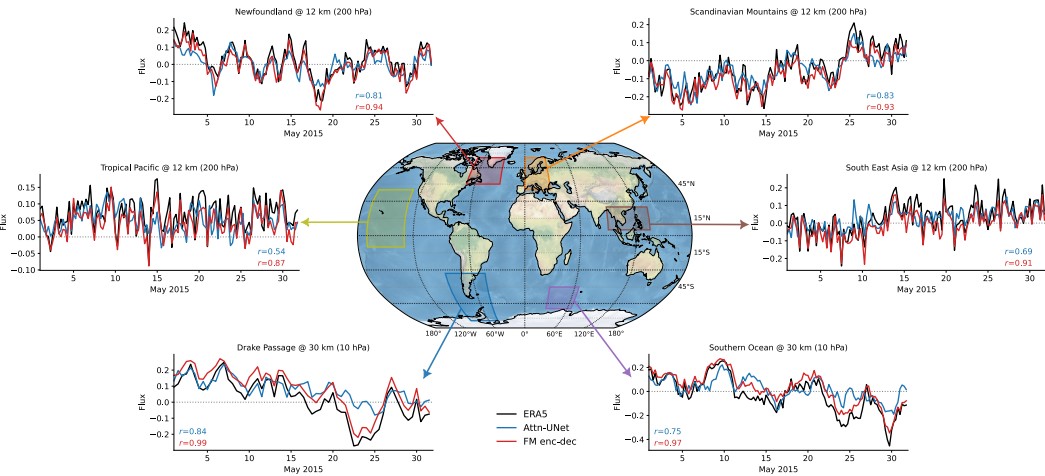

Figure 7: Instantaneous fluxes for May 2015 from ERA5 (black), and predictions from the baseline (blue) and the fine-tuned model (red) over six different hotspots. The numbers show the respective Pearson correlation coefficient w.r.t. ERA5 timeseries. The fluxes in the winter hemisphere are shown at 30 km, but the fluxes in the summer hemisphere are shown at 12 km, since GW activity in the summer at 30 km is substantially weaker.

Based on previously documented studies (Hindley et al., 2020; Wei et al., 2022), the time evolution of the box-averaged fluxes is analyzed for May 2015 over 6 well-known hotspots of GWs (Figure 7).

We are interested in evaluating the nonlocal propagation of GWs as well, which is more prominent in the winter stratosphere (Sato et al., 2012; Gupta et al., 2024a), so wherever possible, we assess and show the transient evolution in the upper winter (southern) stratosphere (10 hPa $\sim$ 30 km). For regions in the summer (northern) hemisphere, we instead analyze the fluxes in the upper troposphere (200 hPa $\sim$ 12 km) instead.

The fine-tuned model generates significantly better prediction for all six hotspots. Most notably for Andes (mountain waves) and the Southern Ocean (non-mountain waves), the predictions from the fine-tuned models bear a correlation coefficient (with the observed fluxes) of 0.99 and 0.97 respectively. In comparison, the respective correlations for the Attention U-Net baseline are 0.84 and 0.75. The correlation with observations is weakest over the Pacific when the fine-tuned and the baseline model predictions bear a correlation of 0.87 and 0.54 respectively.

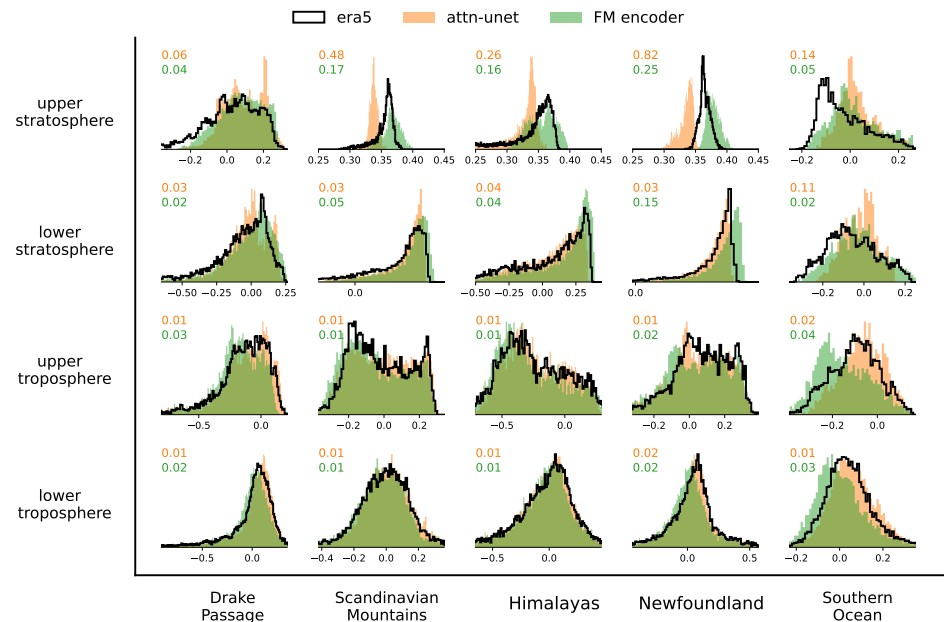

Figure 8: Global flux distributions similar to Fig 6 but segregated according to hotspots. Unlike Fig 5, the figure shows fluxes averaged over boxes outlined in Fig 7.

The successful prediction of both the quick bursts of flux intensification in the tropics (from tropical storms and convective systems) and the slower flux intensification in the midlatitudes (from mountains and storm tracks) demonstrates the fine-tuned model's ability to learn the intermittent and nonlocal evolution of medium-to-small scale atmospheric variability. This is further corroborated by the spatial structure of the predicted flux in Figure 1. Likewise, the finetuned model generates both a richer and more accurate variability in the stratosphere than the baseline (Figure 11).

Revisiting the spatially segregated spectrum of Figure 6, this time exclusively for GW hotspots, we find something similar: the spectrum generated by the two ML models are not so different in the troposphere and lower stratosphere, but the spectrum generated by the fine-tuned model in the upper stratosphere is substantially better than that by the baseline (the only exception is over Newfoundland in the lower stratosphere).

## 4 DISCUSSION

The application of foundation models in climate science is largely unexplored. The analysis presented here clearly establishes that the atmospheric evolution learned by large, transformer-based architectures, can be leveraged to simplify, improve, and expedite the creation of physical parameterizations for climate models; ultimately improving climate prediction accuracy. From a machine learning perspective, since a foundation model (here Prithvi WxC) is typically trained on large

amounts of data, its latent encoder-decoder space contains a rich abstract representation of the atmospheric evolution. In the case of Prithvi WxC, the data ranges from winds, humidity, and radiation, to even leaf area index and soil moisture. Rather than creating task-specific ANNs, CNNs, etc. from scratch, we showed that pre-trained encoders from weather foundation models can instead be leveraged to create better predictive models for atmospheric-oceanic processes. As an added benefit, this approach allows blending data from multiple streams — synthetic high-resolution model data, satellite trains, terrestrial remote sensing data, ground observations, etc.

Our fine-tuned model learns a new atmospheric process, gravity waves, from a totally different dataset, and clearly outperforms the Attention U-Net baseline on all three metrics. In doing so, we also devised a new metric – the tail-Hellinger distance – which allows focusing explicitly on the distribution tails. More specifically, our fine-tuned model both learns the three-dimensional evolution of GWs in the atmosphere and generalizes to regions unseen during training. As a result, this model can arguably represent the missing gravity wave physics in coarse-climate models better than traditional single-column physical parameterizations. This provides further incentive to develop data-driven parameterizations for other parameterized processes like clouds and precipitaion.

## 4.1 LIMITATIONS AND FUTURE WORK

First, skillful offline performance does not necessarily equate to skillful online performance (Brenowitz et al., 2020). Thus, efforts are under way to couple our fine-tuned scheme to a coarse-climate model and assess its online performance and speed.

Second, numerical climate models, climate reanalyses, and data assimilation systems can have systematic biases. Training fine-tuned models on a given foundation model thus presents the danger of the inherent biases in training data to be carried over to the fine-tuned model. Such potential dangers can be alleviated by (a) using multiple data streams, for e.g., using data from multiple climate reanalyses, (b) by combining high-resolution climate data from models with multiple underlying numerics (spectral, finite-volume, spectral element, etc.), (c) using climate model data from multiple climate-change scenarios, (d) by using high-quality data during fine-tuning, and (e) using a probabilistic ensemble of initialization to quantify predictive uncertainty. As a straightforward test, the encoders and decoders from other large AI weather forecasting models can be used to develop and intercompare a series of fine-tuned climate model parameterizations (work in progress).

Third, ERA5 still misses a substantial portion of atmospheric gravity waves (with wavelengths shorter than 150-200 km), so our fine-tuning data is certainly not of the highest quality. This will be improved in future work where the fine-tuning will be accomplished using kilometer-scale high-resolution models instead. Finally, while our fine-tuned scheme generalizes well to regions unseen during pre-training, limited fine-tuning still questions its generalizability to future-climate scenarios. This is yet to be tested and is left for future work.

## 4.2 BROADER IMPACT

Foundation models open avenues to using multi-source observations to facilitate AI-powered climate research. Due to constraints on computing power, we are still decades away from being able to run climate models (such as those participating in CMIP) multiple decades and centuries into the future at kilometer or sub-kilometer resolutions. This means climate prediction will continue to miss crucial sub-grid physics and will continue to rely on physical parameterizations of unresolved processes. This prohibits effective climate action and decision-making and also limits a complete mechanistic understanding of our climate. We have demonstrated a strong application of an existing weather and climate foundation model to climate model improvement. Ideally, foundation models like Prithvi WxC can be used for a whole spectrum of other climate applications.

Since the fine-tuned models can be less costly to train than the baselines — because only a fraction of parameters are retrained — this approach also has the potential to reduce the carbon emissions associated with the from-scratch training of task-specific ML emulators. Used alongside other foundation models, like Prithvi HLS (Jakubik et al., 2023), one can use this approach to even create lightweight, finetuned models for key applications like wildfire prediction, predicting hurricane storm surges, and regional heat wave impacts, potentially improving extreme climate prediction and climate change preparedness.

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

## A  APPENDIX

### A.1  REGIONS

For regional flux analysis, the poles are defined as latitudes 60°-90°, the midlatitudes are defined as 30°-60°, and the tropics are defined as 15°S-15°N.

### A.2  THE TAIL-HELLINGER DISTANCE

We devised a new metric to compare the distribution tails, the tail-Hellinger distance, as follows:

Assume two distributions, $p$ and $q$, with $q$ being the true underlying distribution and $p$ being the predicted distribution. The tail-Hellinger distribution metric most effectively captures the differences in the tails regions if:

(a) the distance of $p$ is computed w.r.t. the underlying prior $q$. Unlike the Hellinger distance, the tail-Helliger distance is not symmetric.

(b) the two distributions are nearly identical in the bulk.

With $q(x)$ defined on $x \in \mathbb{R}$ as the true distribution, let $\mathcal{V} = (\infty, x_1] \cup [x_2, \infty)$ be the tail subset of $\mathbb{R}$, such that for the cumulative distribution function $F$ of $q$, $F(x_1) = \epsilon$ and $F(x_2) = 1 - \epsilon$. That is, $\mathcal{V}$ captures the extremes of the distribution for the prescribed tolerance $\epsilon$. Thus, by definition, $\int_{\mathcal{V}} q(x)dx = 2\epsilon$. In this study, $\epsilon = 0.025$.

Recomputing the Hellinger distance for this interval:

$$\frac{1}{2}\int_{x \in \mathcal{V}} \left(\sqrt{(p(x)} - \sqrt{q(x)}\right)^2 dx \;=\; \frac{1}{2}\int_{\mathcal{V}} p(x)dx + \frac{1}{2}\int_{\mathcal{V}} q(x)dx - \int_{\mathcal{V}} \sqrt{p(x)q(x)}dx \quad (7)$$

$$=\; \epsilon + \frac{1}{2}\int_{\mathcal{V}} p(x)dx - \int_{\mathcal{V}} \sqrt{p(x)q(x)}dx. \quad (8)$$

$\epsilon$ is preferrably small to focus on the tails. Since the integrals are $\ll 1$ for small values of $\epsilon$, we normalize the whole integral by $2\epsilon$, to get:

$$\mathcal{H}_{T,\epsilon}(p,q) \;=\; \frac{1}{2\epsilon}\left(\frac{1}{2}\int_{x \in \mathcal{V}} \left(\sqrt{(p(x)} - \sqrt{q(x)}\right)^2 dx\right) \quad (9)$$

$$=\; \frac{1}{2\epsilon}\left(\epsilon + \frac{1}{2}\int_{\mathcal{V}} p(x)dx - \int_{\mathcal{V}} \sqrt{p(x)q(x)}dx\right) \quad (10)$$

$$=\; \frac{1}{2} + \frac{1}{4\epsilon}\int_{\mathcal{V}} p(x)dx - \frac{1}{2\epsilon}\int_{\mathcal{V}} \sqrt{p(x)q(x)}dx, \quad (11)$$

which is the expression in Eqn 6. For $\epsilon = 0.5$, the whole distribution is considered, thus, the tail-Hellinger distance simplifies to the standard Hellinger distance. Similar bulks, allow focusing exclusively on the tails. Hence, tail-Hellinger distance might not be very informative if the bulk of the distributions are notably different - either in shape (as happens in Figures 5 (right) and 10 (right)), or due to constant offsets in mean.

For very similar tails of $p$ and $q$, the tail-Hellinger distance will be close to zero. If $p$ has a fatter tail than $q$ on average, the tail-Hellinger distance will be positive and vice versa. For predictive distributions $p$ that fail to capture the range of the true distribution $q$ (lower variance), but are similar in the bulk, the tail-Hellinger distance can be expected to converge to 0.5.

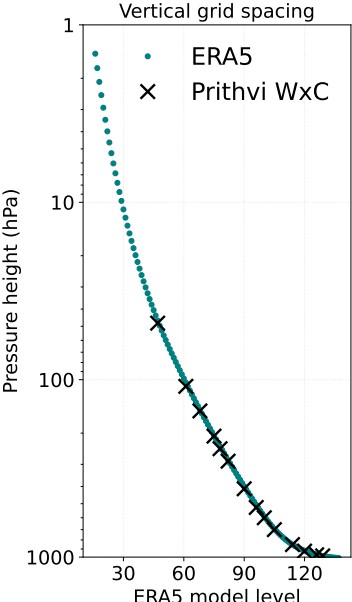

Figure 9: Prithvi WxC was pre-trained on much sparse data in the vertical. The ERA5 fine-tuned data was computed on 137 model levels and the top 15 model levels (i.e. levels above 1 hPa $\sim$ 45 km) were discarded due to an artificial model sponge imposed at those levels. So, effectively 122 model levels between 1000 hPa (surface) to 1 hPa (45 km) height. In contrast, Prithvi WxC is trained on MERRA-2 data interpolated to 14 vertical levels: [985, 970, 925, 850, 700, 600, 525, 412, 288, 245, 208, 150, 109, 48] hPa. No training data between 50 hPa and 1 hPa was provided during pre-training. This means that the frozen encoder-decoder do not have any prior knowledge about the dynamical evolution of gravity waves at these heights. Still, as the analysis shows, the fine-tuned model outperforms the baseline in this region.

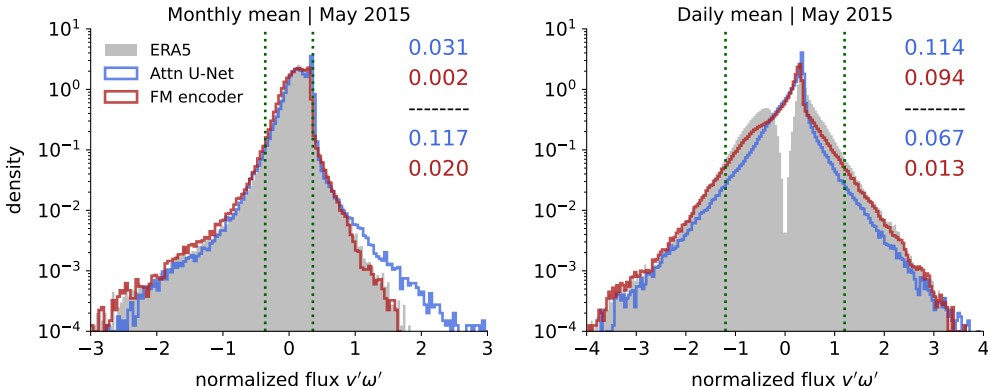

Figure 10: Same as Figure 5 but for the meridional flux $v'\omega'$. The distribution of the (left) May 2015 averaged and (right) daily averaged GW flux $v'\omega'$. The differences around the tail between the Attention U-Net and the finetuned model are more striking for $v'\omega'$ than for $u'\omega'$. The two distributions on the left have an almost zero Hellinger distance. However, the tail-Hellinger distance of 0.117 and 0.02 better captures the difference between the two tails. Similarly for the

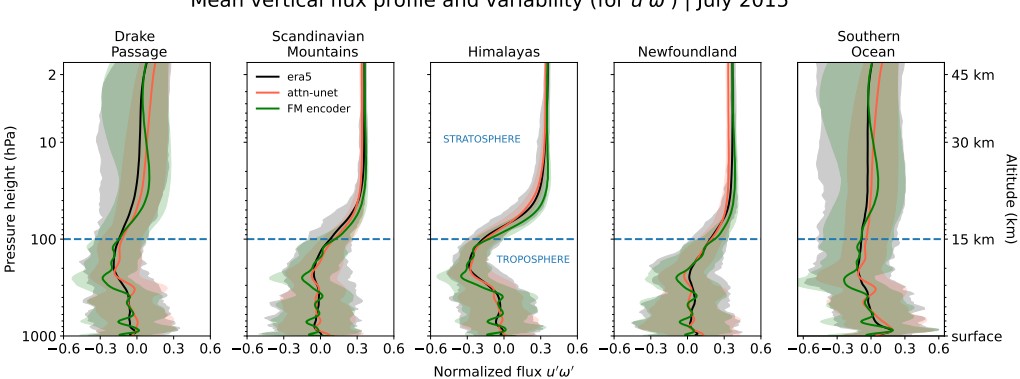

Figure 11: May 2015 mean true and prediction vertical profiles of the zonal flux, $F_x = u'\omega'$ over 5 hotspots. The exact boundaries of the hotspots are shown in Fig 7. The true (but normalized) flux from ERA5 is shown in black, the prediction from Attention U-Net baseline is shown in orange, and the prediction from the fine-tuned model is shown in green. The gray, orange, and green shadings show the range of flux variability in the respective models. The variability is weak in the stratosphere for the Scandinavian Mountains, Himalayas, and Newfoundland because they all lie in the Northern Hemisphere (summer hemisphere for May 2015). Otherwise, the variability over Drake Passage and Southern Ocean is strong. The variability generated by the fine-tuned model (green shading) agrees more strongly with the variability in ERA5 (gray shading), than does the Attention U-Net baseline (orange shading).

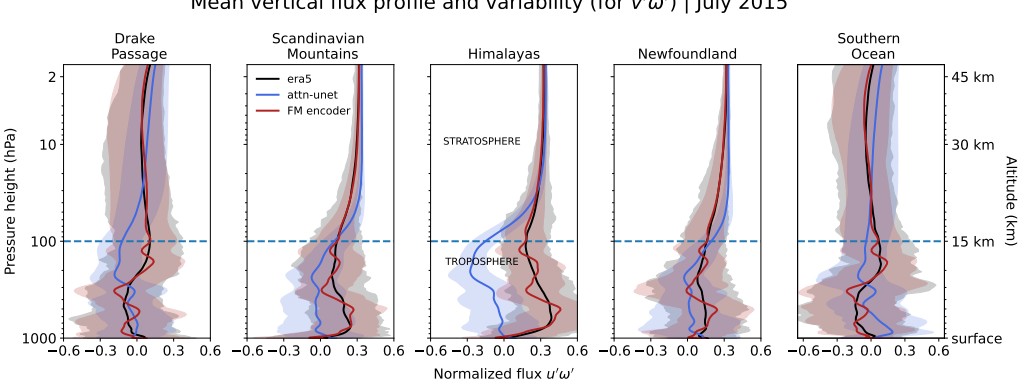

Figure 12: Same as Figure 11 but for the meridional flux $v'\omega'$.

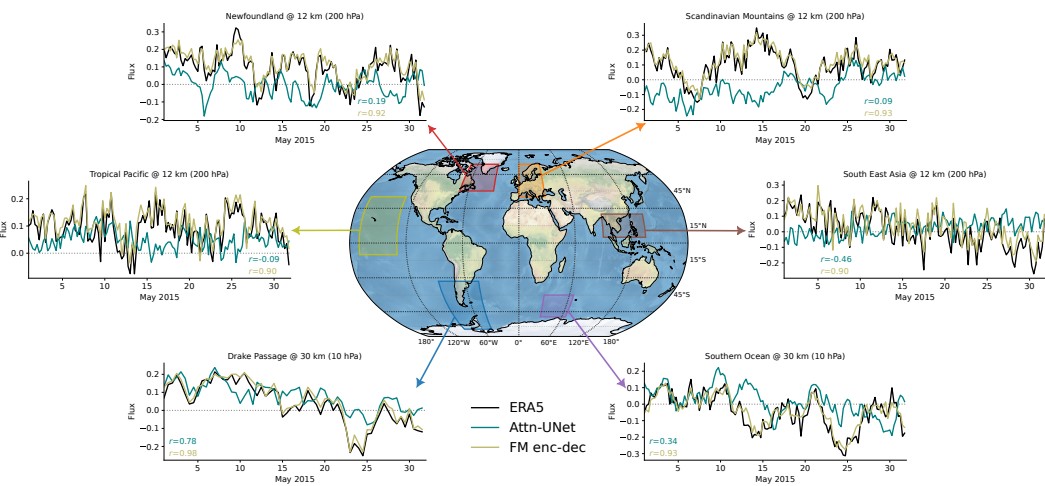

Figure 13: Same as Figure 7 but for $v'\omega'$ - instantaneous fluxes for May 2015 from ERA5 (black), and predictions from the baseline (teal) and the fine-tuned model (light green) over six different hotspots.

