# OpenReview forum: "Finetuning Weather Foundation Models to Develop Climate Model Parameterizations"
_ICLR.cc/2025/Conference — ICLR 2025 Conference Withdrawn Submission_

### Official Review · Reviewer_DYbv · 2024-10-31

**Soundness:** 1
**Presentation:** 3
**Contribution:** 2
**Rating:** 3
**Confidence:** 4

**Summary:**

A recent foundation model for weather and climate (Prithvi WxC) is fine tuned to predict gravity waves. Those predictions are supposed to parameterize subscale processes of coarse climate models and thus improve climate projections. The fine tuned model outperforms a single-task U-Net and generalizes well in regions that are outside Prithvi WxC's training range.

**Strengths:**

_Originality:_ The proposed manuscript touches on relevant topics of the society and applies state-of-the-art foundation models. It is great to see that the fine tuned foundation model generates accurate predictions even outside the regime of the foundation model's training data.

_Clarity:_ The manuscript is mostly clear and well organized. Supplementary material and code for replicating the results are not provided, though.

**Weaknesses:**

_Quality_
The quality of the manuscript lacks in several aspects. Mostly, the comparison of the foundation model vs the baseline seems unfair.
1. Unclear for what reason the authors choose Prithvi WxC as foundation model and not ClimaX or AtmoRep. Comparing these three foundation models can be considered more fair than comparing the fine tuned FM against conventional one-task approaches, like Attention U-Net etc.
2. In the same vein, given that the baseline and fine tuning models are trained on different sets of input variables (lines 203 through 210), it is unclear how to differentiate between the model quality and the data selection.
3. The baseline is a convolutional architecture, whereas the fine tuned model is a transformer, which questions the role of the pretraining vs. the model architecture. It would be great to see how a transformer would compare as baseline model. Similarly, the number of parameters of the U-Net (35M) is not comparable to that of the fine tuned model (2.3B).
4. A plot showing the convergence of the baseline vs the fine tuning model would be informative to better capture their behavior (line 271).
5. In Figure 5, the dotted lines are described to indicate the 2.5th and 97.5th percentiles. The data distributions, however, hardly confirm this. There appears to be substantially more than 2.5 percent of the data left and right of the blue dashed vertical lines.
6. How is the argument in line 325 substantiated, that a Hellinger distance of 0.05 or less is considered pretty good? Is there some literature or data that suggests this decision?

_Significance_
It is hard to assess whether the proposed method is relevant for climate forecasting, mostly as the parametrizations are not tested in numerical climate models.
1. Output fluxes are on fairly coarse resolution and I'm concerned that the coars gravity wave precitions are of limited value for a numerical climate model?
2. The model is introduced to provide parametrizations for climate models, however the study does not test those parametrizations in climate models. As detailed in the limitations section, the actual verification of the parametrization is a key contribution that I consider substantial for assessing the relevance of the proposed method.


_Minor comments_
- lines 37-38: Add details about why SOTA future climate projections are highly uncertain
- line 289: Typo in "evolutionlution"

**Questions:**

1. What spatial and temporal resolution is required to resolve gravity waves? (see abstract and line 60) Please add details to the manuscript.
2. In the caption of Figure 2, what does it mean that encoder and decoder blocks are frozen and used for fine tuning? This reads conflicting, since frozen weights cannot be fine tuned. Also, this figure does not seem to convey much information for the model setup at hand. I have difficulties extracting details about data or architecture. EDIT: In Section 2.4 this is outlined clearly; I suggest to remove Figure 2.
3. What do the models learn effectively? It seems like the models are trained to approximate Equations (1)-(3) and I do not understand why this is done with deep learning models instead of using these equations directly.

---

> ### Author Response · Authors · 2024-11-21
> **Response to comments on summary and weaknesses**
>
> We thank you for your thoughtful comments. Most points raised are not actually weaknesses and we explain why:
>
> 1. **Choice of Foundation Model**: Aurora’s stable code was publicly released on August 22 2024. By then we had already completed our analysis and started writing the manuscript, so it made little sense to go back and redo the analysis just because there is a new foundation model in town. Also, it seems like Aurora pretty much focuses on weather forecasting and air quality forecasting as the only downstream task, so science-wise there's not much difference between Aurora and Prithvi. AtmoRep does not discuss any downstream tasks whatsoever in their non-peer reviewed paper. If the authors themselves have not discussed many downstream tasks in their paper, it makes us a bit skeptical using the model for our analysis as well. ClimaX is too coarse resolution and trained on potentially-biased CMIP6 model output to serve as a meaningful/good choice for climate science applications. Thus, we find Prithvi WxC to be more balanced in terms of both pre-training and fine-tuning applications. Despite not using the models, we reiterate that intercomparing the foundations models is simply not the focus of this study. That said, the goal is also not to suggest that Prithvi is the optimal foundation model for the task of climate model parameterization development. The goal is to open the field of climate science/modeling open to the broader AI community by devising novel use cases in the domain and providing a robust roadmap or proof-of-concept. In principle, any well-designed FM could be used for this ML parameterization development task. We just demonstrate it using Prithvi since it was developed by a mix of both ML and climate-science domain experts and has a well-defined encoder-decoder architecture.
> 2. **Features and data selection**: we should clarify that using different input variables for the two models does not create any discrepancies whatsoever because (as also mentioned in the manuscript) the potential temperature (𝛉) used as input for one model is simply a product of temperature (T) and pressure (p): 𝛉 = T*(p)$^{constant}$. Thus, both models contain the same amount of physical information, just in different dimensions, and these models are robust enough to not be affected by these differences. We like your point though, and we would be happy to add the loss curves for model training using different feature combinations in the revised manuscript.
> 3. **Choice of baseline**: we used an Attention UNet as a baseline because it is a state-of-the-art method for dense prediction. Even previous papers like ClimaX compared their forecasting performance to a UNet and ResNet (though not time series). While we are not comparing our study to theirs, we have certainly drawn inspiration from their approach. Also, since Attention UNet has been accepted well by the ML community, we are confident it can serve as a very effective baseline for our instantaneous mapping task. The fine-tuned model is definitely using 2.3B parameters, but to note, we have frozen the weights of the model so these parameters are not changing, only trainable parameters are the convolution around the Prithvi WxC. Sure, we can definitely go ahead and train a transformer for it, but we also want to acknowledge and build on the accepted published baselines for this task (the Gupta et al. (2024), ICML paper, https://arxiv.org/pdf/2406.14775)
> 4. Sure, we would be happy to share it in the revised version.
> 5. **Log-scaling**: Please note the log scaling of the y-axis. The dotted lines are indeed the 2.5th and 97.5th percentile (see linear plot here: https://tinyurl.com/iclr25)
> 6. **Hellinger distance**: unfortunately, we could not find a super-relevant study which could act as a reference here to interpret the exact values of the hellinger distance. However, our decision to use 0.05 is explained and supported by monte carlo Gaussian sampling conducted over thousands of Gaussian samples, where we found that a Hellinger of 0.05 for standard Gaussians is explained by a roughly equal, i.e.,~60% perturbation in the mean or a ~60% perturbation of the std. dev. This also makes interpreting results easier in terms of increased spread or increased shift in the mean, i.e. if the shapes are similar, the Hellinger distance  above 0.05 can be interpreted as being large due to changes in standard deviation or due to changes in standard deviation. Would be happy to elaborate further in the revised manuscript supported with appropriate figures. Since the Hellinger distance ranges from 0 (identical) to 1 (disjoint), a distance of 0.05 definitely ranges on the lower side indicating the distributions are indeed strongly similar with minimal divergence. When compared to other measures (e.g., KL divergence, Wasserstein distance, etc.), 0.05 often aligns with very small discrepancies. In high-precision domains, a Hellinger distance of 0.05 might be considered excellent.

---

> > ### Author Response · Authors · 2024-11-21
> > **Comment on code availability**
> >
> > We should note that the code is already available on github and model weights are available on Hugging Face, but were not shared to ensure anonymity. A link to both will definitely be added in the final manuscript.

---

> ### Author Response · Authors · 2024-11-21
> **Response to Significance**
>
> Thank you for your questions. There clearly seems to be some confusion and we clarify it below. We will also add these clarifications to the final revised manuscript.
>
> 1. We completely disagree here because our choice to coarsegrain the fluxes is based on robust scientific principles and **utmost care was ensured while creating the model training data**. We arrived at the decision only after consulting multiple domain experts on gravity waves. We should clarify that the momentum fluxes used to the train the models are conservatively coarsegrained (and not simply interpolated) using Python's xESMF library, so they preserve all the information from the 25 km fine-grid by averaging along the longest-resolved wavelengths. Even if we were using a 1 km climate model output, the correct way to define the fluxes would be to coarse grain them onto a coarser grid appropriately chosen according to the longest resolved wavelength, otherwise, we would have to deal with ringing effects associated with wave phases. **Simply put, even as the final grid seems coarse, it preserves all the knowledge from the high-resolution dataset and contains all the relevant information for climate models**. All this would lead to high errors in ML-based predictions and would be rendered fruitless for climate science applications. Since climate models are typically coarse, the coarse resolution selected here (100-300 km) is exactly the optimal fit. Too fine of a resolution would mean that we are predicting wave phases - which would be a poorly-defined problem. In this 'coarsegrained' form, the fluxes could be used optimally by climate models as the traditional parameterizations too use a similar approach to compute momentum fluxes.
>
> 2. Yes, efforts are underway to test the online performance of this scheme. We understand the importance of online testing and have clearly acknowledged in the conclusion section that we are working on it. Since climate models are complex fortran codes, coupling the torch model to a climate model is a challenging technical problem - especially if we want the coupled ML model to provide the optimal speed (which we do). We had to overcome some key logistics problem to build a team that can effectively help us with this and we are now making quick progress on this. Moreover, most climate model parameterizations are tested offline first. **So, this does not reduce the significance of this work at all as rigorous offline testing (the three proposed tests) is key to ensuring stable online performance later**. Also, we appreciate that this may not have been clear from the text but we are not “forecasting climate” here, rather representing missing physics in climate models at any given instance by learning from high-resolution reanalysis data. Otherwise, this line of thought would seem to suggest that most downstream tasks proposed in foundation models like ClimaX, Aurora, or Prithvi have no value because they are offline; this is clearly not the case. While our downstream tasks are not online yet, its rigorous offline testing has inherent value to ensure optimal online performance and the analysis presented here comprises more than half of the entire problem.

---

> > ### Author Response · Authors · 2024-11-21
> > **Response to Questions**
> >
> > Thanks for the three questions. We clarify the concerns below.
> >
> > 1. **Resolution:** ideally, a spatial resolution of 500 m to 1 km and a temporal resolution of 5-10 minutes should be sufficient to resolve most gravity waves (GWs) in the atmosphere. However, datasets that follow both criteria are quite limited as (i) models provide high-resolution in space but not in time due to memory factors. The state of the art global 1 km climate model output provides atmospheric data for 8 months but only every 3 hours, and the whole data takes more than 3 petabytes of space. On the other hand, satellites & balloons provide high-frequency data, but have poor spatial resolution. Thus, ERA5 currently provides the best dataset/trade-off between spatial (25 km) and temporal (hourly) resolution. We will make sure to elaborate more on this in the revised manuscripts.
> >
> > 2. **Frozen encoder-decoder:** Finetuning can be done in several ways where we can freeze all layers/some layers or none, and add couple of trainable parameters in the beginning and end of the model. Here is our thought process: if we load the model for finetuning, we would have to adopt the FSDP approach so that model layers can be finetuned, which will need a minimum of 4 A100 40 GB around fine-tuning. However, freezing the model helped us to load it on a single GPU. This works great to ensure a wider applicability by a broader community of ML scientists and climate scientists who may have limited compute resources to create such parameterizations. Additionally, making all the layers trainable in the decoders would not have offered improvements as the model was pretrained on 14 vertical levels from MERRA 2 and finetuned on 122 vertical levels from ERA5. Considering both of the dataset have different assimilation strategies from raw observations, the model will need to learn the relationship between both MERRA5 and ERA5, which might differ by parametrization/handling of different PDEs (assimilation objectives, finite vs pseudospectral numerical methods). So, we froze the model and added convolution in the beginning, which would have learned local features from the data, and then hit the embedding space. One can call this “local information injection”. So the model weights are changing but not of base model but of surrounding convolutions. Thanks for the suggestion though, and letting us know that this was not clear at first from the text. We will improve the caption and if needed, move the figure to the Appendix in the final manuscript.
> >
> > 3. **Learning closed-form expressions:** Good question! the underlying closed forms are approximately known from linear wave theory. But low resolution climate models cannot meaningfully resolve these terms because a bulk of the contribution in these terms come from spatial scales 100 km and shorter, and these scales are not resolved in a typical climate (not weather) model . So, we extract these quantities from high-res observations (which represent physics) to prepare training data and then use the trained ML model to represent these missing physics/unresolved terms in low-res climate models, boosting their physics representation. So, in short, we know what physics climate models are missing and why - so we resort to high-resolution climate data/observations to learn the physics using AI and analytical forms - and then we coupled these ML models back to the low-res climate models to represent the missing physics.

---

### Official Review · Reviewer_XXRg · 2024-11-03

**Soundness:** 3
**Presentation:** 3
**Contribution:** 2
**Rating:** 5
**Confidence:** 4

**Summary:**

This paper presents a data-driven parameterization scheme for gravity waves, aiming to achieve gravity wave parameterization in coarse climate models by fine-tuning weather foundation models. The work utilizes the newly proposed Prithvi WxC as a pre-trained model and leverages higher-resolution ERA5 data to achieve this goal. Their main contribution is both reducing the costs of training data-driven parameterization models from scratch and improving their generalization capabilities through fine-tuning foundational models.

**Strengths:**

1. Significance: This work introduces a fine-tuning algorithm for weather foundation models into parameterization, achieving a more lightweight and better generalization performance data-driven parameterization scheme.
2. The manuscript is well-written and comprehensible, adhering to ICLR formatting guidelines, with no discernible errors.
3. The performance of the proposed algorithm is demonstrated through three different evaluation methods.

**Weaknesses:**

1.**Innovativeness:** This work showcases commendable application performance on specific tasks; however, its contribution to the broader machine learning community may be viewed as somewhat limited. Additionally, the fine-tuning scheme is discussed in detail in Section 3.2 of the cited paper that presents the pre-trained foundational model used here, where it is described as a typical downstream task. Given the context of a high-level conference, the innovativeness of this work might be considered subtle.

2. **Experimental Setup:** It seems that neither the methods relying on equations nor the mixed probability methods mentioned in the 'Related Work' section are included in the baselines; rather, the comparison is made with Attention U-Net. Furthermore, the input variables for Attention U-Net are one-quarter fewer than those utilized in this work, resulting in a reduction of input information. Could you clarify why it is not possible to fully input the data of [488, 64, 128]?

3. **Experiments:** In the daily average GW momentum flux experiments (Figures 5 and 10), notable differences are observed in the distribution of small-scale features between the deep learning model and ERA5. It would be beneficial to include the MERRA-2 distribution to clarify that the data-driven parameterization scheme successfully learns GW information from the ERA5 dataset, rather than simply inheriting features from a pre-trained model.

**Questions:**

1. In the "Contributions" section, the authors suggest that this method could potentially be extended to cloud parameterization and precipitation forecasting (line 105). However, these topics are not explored within this work. Given their distinct nature compared to gravity wave parameterization, and considering that the ERA5 dataset may lack reliable data for these variables, could the authors provide additional experiments or insights to substantiate this claim?

2. In lines 201 and 202, the authors state, "This corresponds to roughly 35k training samples, which pretty much classifies as 'data-scarce'." This definition of sample size seems to differ from what is typically observed in other areas of the machine learning community. Could the authors provide context on the typical dataset sizes used for similar climate modeling tasks, and explain why 35k samples is considered data-scarce in this domain compared to other machine learning applications?

3. Could the authors clarify how the fine-tuning and test datasets are partitioned for the experiments? Specifically, please provide the exact split ratios for training, validation, and testing, and indicate whether the data was split randomly or using another method, such as temporal splits for time series data. This would enhance readers' understanding of the experimental setup.

---

> ### Author Response · Authors · 2024-11-20
> **We clarify why the comments do not classify as weaknesses**
>
> Thankyou so much for your thoughtful comments. We clarify why the concerns do not really classify as weaknesses.
>
> # Weaknesses:
>
> 1. **Innovativeness:** The use of AI to empower climate modeling (not weather forecasting) has been severely limited. Our work opens avenues for the broader ML community to develop AI models to advance climate modeling and climate science (which has a different set of challenges than weather forecasting). We also introduce the tail-Hellinger distance, a novel metric focusing on the accuracy of the predicted tails—crucial for capturing rare but impactful events in any domain where ML is applied to learn distributions. Our work bridges ML advancements and high-impact applications in climate science (not weather), promoting interdisciplinary research and deployment of FMs in other scientific fields, and provides a pathway for ML experts to foray into climate science. It is worthwhile mentioning ClimSim - which won the best paper at NeurIPS 2023. ClimSim did not introduce any new model architecture, but provided a dataset/pathway to allow ML experts to work with ideas in climate modeling. We have appropriately submitted our work to the “Applications to physical sciences” area of the conference.
>
>
> 2. **Experimental Setup: We disagree here**. The two methods have been used in different contexts. Eqn disc has been used for momentum closures in the ocean (where analytic forms are not known) and the prob. models focus on combining low-fidelity and multi-fidelity datasets for precipitation. **Both setups also require different datasets than ours**. Moreover, in our case, the analytic forms are already known but not resolved.  We searched the literature and found only one study which has looked at (resolved) gravity wave fluxes (Gupta et al. (2024); https://arxiv.org/pdf/2406.14775). Subsequently, here we used their Attention UNet baseline. Using different inputs for the two models **does not create any discrepancies** because (as also mentioned in the manuscript) the potential temperature (𝛉) is simply a product of temperature (T) and pressure (p): 𝛉 = T*(p)$^{constant}$. So, both models have the same physical info, in different dimensions, and these models are robust enough to be unaffected by these differences.
>
>
> 3. **Experiments:** Thanks for the comment but **this is definitely not the case (and does not qualify as a weakness)**. MERRA-2 has (a) a factor two coarser grid (0.5 deg x 0.625 deg as opposed to ERA5’s 0.25 deg) and (b) has finite volume numerics. As a result, MERRA2 does not provide a competent GW field (Li et al. 2023, https://doi.org/10.1002/qj.4605). Also, including (weak) GW fluxes from MERRA2 won't be informative because the small value prediction affects the UNet baseline and the finetuning model alike. Otherwise, the UNet baseline, which is pretraining agnostic, would have shown better skill in predicting small values. If anything, we see the opposite, i.e., pretraining on one dataset and finetuning on the other leads to improvements wrt baseline.
>
>
> ---
> # Questions:
> 1. A similar strategy can be applied to develop other params e.g. clouds. We do not suggest using ERA5 to develop these since it under-resolves convection. One can use a mix of high-resolution climate model output and satellite irradiances and provide task-specific input data like specific humidity, saturation pressure, precipitation fluxes, latent heat fluxes etc. from other high-res datasets to develop these ML params. The baseline models would also vary. For GWs, we only found one existing baseline — the Attention UNet (Gupta et al.) — to predict the small-scale fluxes, and so we used that to inform our finetuning. Similarly, in the future, SOTA benchmarks for cloud params & pecip. (and other processes) could be used to compare the performance of finetuning models. Alternatively, in case of a lack of a baseline, encoder-decoder pairs from other FMs can promote effective intercomparison of param. architecture.
>
> 2. Data-scarcity. **Past ML param. studies:** Wang et al. (2022)(https://doi.org/10.1029/2022MS002984) use ~4.5 mil samples as training+validation set. Espinosa et al. (2022)(https://doi.org/10.1029/2022GL098174) use ~11 mil samples. Similarly, Zanna and Bolton use 10 yrs of high-res training data. **Temporal coverage:** we (purposely) used only 4 yrs of data for training. This could lead to: (1) underrepresentation of tropical convection and hence gravity waves: processes like the El-Niño Southern Oscillation have a typical period of 2-7 yrs but are not fully represented. Similarly, the quasi biennial oscillations (QBO) in the tropical stratosphere have a period of 28 months. Since we sample our data over 2010, 2012, 2014, 2015, the training data does not cover both phases of the QBO.
>
> 3. Of 48 months, 47 months training + 1 month (May 2015) validation. We also trained the baseline on 3 yrs and tested it on 1 yr and found similar results & losses. Training data was randomly shuffled.

---

> ### Comment · Reviewer_XXRg · 2024-12-03
>
> Thank you for your detailed responses. Your clarification regarding the "weaknesses," especially the explanation of the "experiments," has addressed my concerns about the completeness of the work. However, I remain uncertain about the paper’s contribution to interdisciplinary research, particularly with regard to its innovative aspects. While this is certainly a strong application of machine learning to climate modeling and forecasting, from a research paper’s perspective, it is essential to emphasize the novelty or irreplaceability of the method.
>
> The reason ClimSim received recognition at NeurIPS 2023 is due to the importance of quality datasets in advancing machine learning, much like the role ImageNet has played in computer vision. While excellent algorithmic applications can have a similar impact, this paper does not sufficiently highlight the unique contributions it makes to the field, community, or society at large. Although I understand the authors' frustrations, I believe that, as an interdisciplinary piece, the paper should more clearly address the varying expectations across fields.
>
> In particular, for a submission to a leading machine learning conference, the paper should be structured around the interests of that community. For instance, it would be beneficial to clearly explain the reasons behind the "data scarcity" issue (thank you for addressing this in your response), and to discuss the dataset split—whether using only 2% of the data for testing may affect generalizability.
>
> Based on these points, I will not change my score, unfortunately. I would recommend considering submission to the "Applied Data Science Track" at a top-tier machine learning conference or to a high-impact journal in the climate science field.

---

### Official Review · Reviewer_ripD · 2024-11-03

**Soundness:** 3
**Presentation:** 3
**Contribution:** 1
**Rating:** 3
**Confidence:** 5

**Summary:**

This paper introduces the fine-tuning of weather/climate foundation models to the task of atmospheric gravity waves.

**Strengths:**

- The presentation of the results is interesting and thought-provoking. Very cool to see the difference in the modeling between baseline and fine-tuned model.

**Weaknesses:**

- The novelty / content of the paper is somewhat limited. For example in ClimaX --  an ICML paper -- which introduced a full end-to-end pretraining and finetuning pipeline over multiple datasets and multiple weather and climate finetuning tasks. A possible extension for this paper would be to use different foundation model backbones - any of the large weather models are interesting here.
- The Prithvi WxC model is very low resolution compared to other models, shows hardly no ablations, and for none of the tasks they consider, they beat any SOTA baselines. The current SOTA in weather and climate modeling has moved way beyond the 0.625x0.5 resolution. This of course is not the fault of the authors. Yet for example, the Aurora foundation model (left out in the paper) is trained on many atmospheric datasets on much finer resolution. It would have been nice to put a comparison between these two models.
- The downstream task is super low resolution - which a priori is ok, but not rtoo impressive.
- At least one more baseline would be needed to really gauge the results.


I would advise the authors to consider user other foundation models too. And please stop refering to the Prithvi WxC model as SOTA model.

**Questions:**

-

---

> ### Author Response · Authors · 2024-11-20
> **Seems like the reviewer might be mixing two very different things here. We clarify.**
>
> We thank the reviewer for putting in valuable time to review our manuscript. It seems like they are mixing two very different things here but that is ok -- we try to answer them below:
>
> - **The goal of this study** is to demonstrate *ONE WAY* in which foundation models (FM)  can be relevant to the domain of AI for climate science modeling (not weather forecasting). ClimaX and Aurora (and even Prithvi) otherwise only address downstream tasks relevant over weekly to monthly timescales (which is weather, not climate!). The study *DOES NOT CLAIM* that Prithvi is the only FM fit to do so. That is simply not the point of this study. And we think the reviewer also understands this which is why they suggest “possible extensions”.
>
> - **Climax has many loose ends: ClimaX – an ICML workshop paper** – was trained on a very coarse dataset of 5 degrees. Even for rollout forecasting the authors used convolution models instead of time series models (UNet and ResNet). For some surprising reasons, ClimaX's authors did not compare their results with (or acknowledge) Fourcastnet, which was the SOTA at that time. However, in this paper we are not creating a comparison between models anyway, rather creating a case for downstream application for climate model (and not weather model) improvement. Given that this is the first study (to our knowledge) to apply FMs to address a physical problem which affects climate (and not weather) models over yearly and multi-year timescales, we strongly disagree that this paper does not have any ML novelty. Prithvi is higher resolution compared to ClimaX and is trained on MERRA2 dataset. We are not presenting Prithvi WxC here, and are merely using it, so we can’t compare its performance for all the tasks. So, we think this concern raised by the reviewer might better belongs to their paper, not ours.
>
>
> - **Aurora code was unavailable before August:** Aurora code was released on August 22. We already finished our analysis and started preparing our paper by then. So, it does not make much sense to  then include a half-cooked analysis in our paper just for the sake of it when it is not needed. Our conclusion stands on its own. This is also consistent with ICLR policy regarding use of fresh work for comparison. But we appreciate that the reviewer might not be aware of this. Aurora's stable code was released on August 22. So its not practical for us to run a completely new model within 2 weeks of submission due date and write down the results.
>
> WP #3
>
> - **This is not correct** - it is *NOT* A "super low resolution" task as you think. The manuscript clearly mentions that the fluxes are “conservatively coarsegrained” and not merely interpolated. This means that the interpolated data contains all the information of the 25 km grid, then projected onto a 300 km grid. Conservatively coarsegraining fluxes is the scientifically accurate way to represent them as the fluxes are only defined in terms of wave averages. The physically accurate way to design a flux prediction experiment is to coarsegrain them to a 100-300 km grid (as that is the scale of the longest gravity waves). This consideration is based on past atmospheric wave dynamics literature (Polichtchouk et al. (2022) for instance; https://doi.org/10.1002/qj.4202). Now, since you mention ClimaX, we should mention that their downstream tasks are actually quite low-resolution because they simply coarsen the input-output pairs to a coarser grid by discarding the finer-scale details. Thus, the task presented here presents a robust test to assess “small-scale” predictions. We appreciate that this subtle detail might not be apparent to reviewer who may be approaching this problem from a pure data-driven perspective.
>
> WP #4
> - **This comment is a bit ambiguous too**, and we strongly disagree with the reviewer. We have used the most recent SOTA baseline to compare our results for gravity waves downstream task. We searched extensively to find the most appropriate baselines for the gravity wave flux task but found only one study (Gupta et al. (2024); ICML 2024- https://arxiv.org/pdf/2406.14775). That study establishes Attention UNet - a SOTA model for dense prediction - as a baseline. We were also inspired by the fact that if ClimaX can use a basic UNet to compare their forecasting results without any temporal component whatsoever, we can definitely use an improved version of their choice to serve as a baseline here.
>
> - Lastly, we feel sorry for the reviewer to be triggered by our reference to Prithvi as SOTA model, even though there is not reason to be. We decided to used Prithvi for our application because it was developed by a team of both ML and climate/weather domain experts. Not sure why the reviewer sounds offensive.

---

> > ### Comment · Reviewer_ripD · 2024-11-20
> > **Thanks for the clarification.**
> >
> > Those answers helped to understand the motivation behind the paper, I agree with the comments. My point about ClimaX was to point out that there are papers which present many downstream tasks, but I understand that your specific downstream task is hard and unique and thus might require a study by itself. I am not raising the score, since I still think the paper needs more meat - I know this is unsatisfying.

---

### Official Review · Reviewer_5F5c · 2024-11-03

**Soundness:** 3
**Presentation:** 3
**Contribution:** 2
**Rating:** 3
**Confidence:** 4

**Summary:**

This paper explores the use of AI foundation models (FMs) to improve climate model parameterizations, specifically focusing on gravity wave (GW) effects. The authors leverage a pre-trained weather foundation model (Prithvi WxC) by fine-tuning its encoder and decoder components to predict gravity wave momentum fluxes that are typically unresolved in coarse-resolution climate models.


The work demonstrates how a foundation model pre-trained on MERRA-2 reanalysis data can be fine-tuned using ERA5 data to create parameterizations that capture gravity wave physics. The authors develop a model that predicts GW momentum fluxes given background atmospheric conditions, comparing their fine-tuned approach against a baseline Attention U-Net model trained from scratch. They evaluate the models using three tests: predicting global flux distributions, region-specific flux spectra across different atmospheric heights, and temporal evolution of fluxes at known gravity wave hotspots.


The results show that the fine-tuned foundation model approach outperforms the baseline, particularly in predicting stratospheric gravity wave behavior - even across pressure levels where the original foundation model was not pre-trained. The authors employ the tail-Hellinger distance to specifically evaluate how well the models capture extreme events in the flux distributions.


The paper positions this work as a proof-of-concept for using foundation models to develop improved climate model parameterizations more broadly, suggesting that similar approaches could be applied to other unresolved processes like clouds and precipitation. The authors argue that this approach offers advantages in terms of training efficiency, generalization capability, and physical consistency, while acknowledging current limitations and areas for future work.

**Strengths:**

The paper puts forth a fine-tuning framework and a rigorous evaluation methodology in applying foundation models to climate science parameterizations. The work's primary strengths lie in its comprehensive experimental design and clear presentation of results. The authors provide a thorough evaluation framework through three increasingly stringent tests - from global distributions to temporal evolution at specific hotspots - which could serve as a valuable template for assessing other climate model parameterizations.


From a technical perspective, the paper employs the tail-Hellinger distance metric for evaluating extreme events in flux distributions, showing careful consideration of evaluation methodology appropriate for climate science applications. The empirical results demonstrate that their fine-tuned approach outperforms a strong baseline (Attention U-Net), particularly in an interesting case where the model generalizes well to stratospheric gravity wave behavior even in regions where the original foundation model wasn't pre-trained.


The paper is very clear and accessible through a well-structured presentation. The authors have done a good job at bridging machine learning and climate science concepts, making the work comprehensible to both communities. The figures are particularly well-designed, with Figure 1 effectively illustrating the gravity wave prediction task and Figures 5-8 systematically presenting the evaluation results across different atmospheric conditions.


While the work doesn't advance machine learning methodology significantly, it provides a well-executed case study demonstrating how foundation models can be leveraged for climate model parameterizations. The successful demonstration with limited fine-tuning data (four years of ERA5) suggests a practical pathway for developing similar parameterizations, though these contributions are more relevant to climate science than machine learning. The clear presentation and thorough validation make the work's climate science implications accessible to a broad audience, even if the core innovations lie primarily in the application domain rather than methodological advancement.


To summarize, I acknowledge the paper's strong execution and clarity while being more explicit about its primary contributions being to climate science rather than machine learning methodology.

**Weaknesses:**

**Technical Innovation:**
The paper's primary limitation lies in its modest contribution to machine learning methodology, which is a crucial consideration for ICLR. While the work presents a compelling application of foundation models to climate science, it essentially applies established fine-tuning techniques to a new domain without introducing significant methodological innovations. The approach largely follows the pre-training/fine-tuning paradigm that has been well-documented in recent literature, including applications in weather and climate modeling [1,2].

**Connection to prior works:**
Similar applications of foundation models in Earth system science have already been demonstrated by works such as ClimaX [1] and Aurora [2], which showed that weather-trained models can be effectively fine-tuned for various downstream tasks, including data-scarce scenarios. The current paper's findings, while valuable for climate science, align with these expected outcomes and don't present surprising methodological insights for the machine learning community.

**Limited contributions to ML:**
From a technical perspective, the paper primarily describes a straightforward application of fine-tuning the Prithvi WxC model to predict gravity wave momentum fluxes. I do acknowledge the engineering effort required to carry out this study, but from a machine learning standpoint I see limited contributions. While the authors have conducted thorough experiments, these contributions are more relevant to climate science evaluation than advancing machine learning methodology. The neural network architecture modifications described in Section 2.4 are relatively standard adaptations rather than novel technical contributions.

**Venue fit:**
The paper's strengths - particularly its thorough evaluation of gravity wave predictions and implications for improving climate model parameterizations - would be better suited for climate science venues where domain experts could properly evaluate the scientific implications. Journals such as Journal of Advances in Modeling Earth Systems (JAMES) or Geophysical Research Letters (GRL) would provide a more appropriate audience and review process for assessing the work's primary contributions to climate modeling.

**Recommendation for improvement:**
While the paper effectively demonstrates the potential of machine learning in climate science, its core innovations lie in the application domain rather than in advancing machine learning methodology. The work would benefit from either substantial enhancement of its machine learning contributions for ICLR or redirection to a venue better aligned with its primary contributions to climate science.

**Impact considerations:**
This assessment isn't meant to diminish the paper's value but rather to highlight that its strengths may be better appreciated and more impactful in a different academic venue. The thorough experimental validation and careful consideration of climate science implications would likely generate more meaningful discussion and follow-up work in the climate modeling community.

[1] Nguyen, T., Brandstetter, J., Kapoor, A., Gupta, J. K., & Grover, A. (2023). ClimaX: A foundation model for weather and climate. arXiv preprint arXiv:2301.10343.

[2] Bodnar, C., Bruinsma, W. P., Lucic, A., Stanley, M., Brandstetter, J., Garvan, P., ... & Perdikaris, P. (2024). Aurora: A foundation model of the atmosphere. arXiv preprint arXiv:2405.13063.

**Questions:**

1. Could the authors elaborate on why they chose to freeze both the encoder and decoder during fine-tuning? Would allowing some layers to be trainable, particularly in the decoder, potentially improve performance? At 2.3 billion parameters, the model size is not too prohibitive for full-parameter fine-tuning.

2. The daily flux predictions show difficulty with small flux values (Figure 5b). Could the authors discuss potential approaches to address this limitation?

3. The tail-Hellinger distance is an interesting metric - could the authors provide more intuition about how to interpret different values, particularly negative ones?

4. Have the authors considered evaluating the model's performance during extreme weather events or seasonal transitions where gravity wave behavior might be particularly challenging to predict?

5. Could the authors provide more details about the data preparation, particularly how the coarse-graining from 25km to 280km resolution was implemented?

6. What was the rationale behind choosing 4 convolutional blocks before and after the frozen encoder-decoder? How sensitive is the model performance to this architectural choice?

7. The authors mention plans to couple their scheme to a coarse-climate model. Could they elaborate on the technical challenges they anticipate in this integration? As climate models require long-time integration of the underlying PDEs do you foresee any stability issues when an ML parametrization is used?

8. The authors chose Prithvi WxC as their foundation model, but recent work has shown Aurora achieving state-of-the-art performance in weather prediction, especially at high resolutions (see [2] above). Could the authors discuss why they selected Prithvi over Aurora, and whether they expect their findings would generalize or potentially improve when using Aurora as the base model? This discussion would be particularly relevant given Aurora's demonstrated superior accuracy in weather forecasting and its potential advantages for learning atmospheric dynamics. It would also help readers understand whether the choice of Prithvi was primarily due to practical considerations (like availability or computational constraints) or if there were specific architectural features that made it more suitable for this particular application.

9. No code was provided as part of the submission. This would help to further assess the technical correctness and reproducibility of the results. Are the authors planning to open source their framework?

**Details Of Ethics Concerns:**

No ethics concerns to report.

---

> ### Author Response · Authors · 2024-11-23
> **Certainly relevant for ICLR**
>
> We thank the reviewer for their detailed comments, and for acknowledging that our work has the potential to be useful to both the ML & the climate science community. However, we do disagree on the degree to which this study could be useful to the broader ML community. Below we explain why.
>
> 1. **Solving climate science use cases can benefit other ML problems too:** An increasing number of AI weather forecasting models and foundation models (FMs) have been developed over the past 3-4 years. The release of FourCastNet [1] inspired both the weather forecasting and machine learning community alike to develop more stable weather forecasting models. Likewise, the release of ClimaX [2] proof-of-concept inspired the development of more versatile FMs like Aurora [3], AtmoRep [4], and Prithvi [5], all of which involved heavy collaboration between the weather forecasting and the AI community. **Yet, the application of these FMs to climate prediction tasks has been severely limited**. Either we wait for these models to be stable over multi-year timescales or we find novel ways to bypass these limitations and use AI to improve climate models (not weather models) now. Literally none of the downstream applications for FMs focus on reducing climate uncertainty – because it typically involves analysis and rollout on longer timescales over which these AI models are not stable, and significant rapid advancements will be needed before they could do so. Our analysis establishes that this should not discourage the use of these models for climate prediction and climate modeling. This is best accomplished by introducing new climate science use cases more accessible to ML experts and by developing new probabilistic metrics potentially useful in broader ML.  **Therefore, the focus of this study is not just to highlight the prowess of FMs for downstream applications (which has been amply done before), but also pioneer a new research direction to use AI to advance climate modeling and climate prediction**, at the same time aligning with societal needs. **Presenting this at ICLR and sharing this with the ML community is the best way to inspire more research in this direction.**
>
> 2. **Submitted to “Application to physical sciences” area:** we totally appreciate that the idea of using a finetuned model to perform weather-related downstream tasks has been put forth by past studies. However, this is the first study to leverage weather-focused FMs to create a viable downstream task for climate model development and the state of the art Attention Unet baseline (state-of-the-art for gravity wave analysis as least) has been used to create a new but similar finetuning model architecture. By doing so, we confident we are redefining the limits of what problems (domain and timescale) weather FMs can be used to address, potentially motivating more climate-focused applications in the future, also creating room for development of better model architectures. In the same spirit, we have submitted our paper to the “application to physical sciences” sub-area of the conference.
>
> 3. **New probabilistic metrics:** The tail Hellinger metric introduced in the study can be used more broadly across different problems, especially in studies which focus on simulating distributions and their tails, like in extreme event analysis and modeling of intermittent systems.
>
> 4. **Relevant audience:** submitting our paper to ICLR therefore ensures that we get the optimal audience whose focus is both on identifying novel avenues to apply ML to advance long-term climate analysis through development of new models and use new metrics to quantify performance of machine learning models to predict intermittent nonlinear dynamics.
>
>
> References:
>
> [1] Pathak, Jaideep, et al. "Fourcastnet: A global data-driven high-resolution weather model using adaptive fourier neural operators." arXiv preprint arXiv:2202.11214 (2022).
>
> [2] Nguyen, T., Brandstetter, J., Kapoor, A., Gupta, J. K., & Grover, A. (2023). ClimaX: A foundation model for weather and climate. arXiv preprint arXiv:2301.10343.
>
> [3] Bodnar, Cristian, et al. "Aurora: A foundation model of the atmosphere." arXiv preprint arXiv:2405.13063 (2024).
>
> [4] Lessig, Christian, et al. "AtmoRep: A stochastic model of atmosphere dynamics using large scale representation learning." arXiv preprint arXiv:2308.13280 (2023).
>
> [5] Schmude, Johannes, et al. "Prithvi WxC: Foundation Model for Weather and Climate." arXiv preprint arXiv:2409.13598 (2024).

---

> > ### Author Response · Authors · 2024-11-23
> > **Response to questions**
> >
> > Thank you for the thoughtful questions. We provide the response below. We provide detailed answers to questions 2 and 3 on daily flux predictions and tail-Hellinger distance separately as a follow-up comment:
> >
> > 1) If we unfreeze and load the full model for finetuning, we would have to adopt the FSDP approach so that model layers can be finetuned, which will need a minimum of 4 A100 40 GB around fine-tuning. However, freezing the model helped us to load it on a single GPU. Additionally, Making some layers trainable in the decoders would not have offered improvements as the model was pretrained on 14 vertical levels from MERRA 2 and finetuned on 122 vertical levels from ERA5. Considering both of the dataset have different assimilation strategies from raw observations, the model will need to learn the relationship between both MERRA5 and ERA5, which might differ by parametrization of different PDEs. So, we froze the model and added convolution in the beginning, which would have learned local features from the data, and then hit the embedding space. One can call this “local information injection”. Later on we use the transformation to reach back to the gravity wave. Considering, change in data resolution- space, time and domain space, we decided to use convolution which would have helped the model to adapt better to the domain.
> >
> > 4. Yes, we plan to test the model performance during (a) seasonal transitions in the southern hemisphere stratosphere, i.e. final warmings, and around extreme events in the northern hemisphere stratosphere, a.k.a, sudden warmings. Getting the gravity waves forcing correct around these features will serve as a strict test of model performance. However, to ensure statistical confidence, this would be best accomplished using online testing. Thus, once our scheme is coupled to the climate model, we plan to test on well-known atmospheric features like final warming dates, sudden warming frequencies, tropical QBO period.
> >
> > 5. Yes, absolutely. The conservative coarsegraining (not the same as linear interpolation) was achieved using the 1st-ord. conservative regridding func. provided by the xESMF Python library. The fluxes were first computed and stored at a 25 km grid, then coarsegrained to a T42 (~300 km) Gaussian grid. We tested different coarsegraining methods in xESMF and found little differences. The 300 km resolution was purposely selected to ensure consistency as we are currently coupling the ML model to a 300 km grid resolution global climate model.
> >
> > 6. Since the Attention UNet baseline also uses 4 downsampling layers (excluding the bottleneck), we selected 4 conv. blocks to strike similarity between the baseline and the finetuning design. Otherwise, it would be susceptible to surmise that performance gains are due to different model depths. We would be happy to provide ablation results in the revised manuscript.
> >
> > 7. Yes, efforts are underway to couple this scheme to a climate model. As mentioned in the paper – good offline performance does not always equate to good online performance. This is due to **nonlinear feedbacks** between the ML scheme and the climate model. Due to such feedbacks, small errors can often grow exponentially leading the model to produce nonsensical results. Moreover, online performance will also present a rigorous test of the ML as a long term simulation will test the **generalizability** of the model to new inputs and new model climatology. One technical challenge in particular is **speed**. Coupling the torch model with a Fortran code and invoking it every model physics step is particularly challenging as the communication leads to an ultimate slowdown of the climate model. We are working towards finding a solution. However, if the scheme is evaluated strictly, as we have attempted to do in this study, it generates sufficient confidence that the scheme can perform plausibly during offline tests as well.
> >
> > 8. We completed our analysis by the last week of August. Aurora’s code was made public on Aug 22 as it made little practical sense to ‘switch’ to Aurora. Also, Prithvi was made by a mix team of both climate/weather scientists and ML scientists, so we stuck to Prithvi. Regardless, since the focus of our study is to open avenues for the application of ML for climate science modeling (not weather science), and since our study is agnostic of the foundation model used, FM choice is not a significant issue.
> >
> > 9. The code is already available on GitHub but we didn’t share it to ensure anonymity. It will be made available in the final version.

---

> ### Author Response · Authors · 2024-11-23
> **Response to questions - part 2**
>
> Here we provide detailed answers to your questions 2 and 3.
>
> 2. Yes, this is an important issue and is reminiscent of the issues common to AI weather forecasting models like FourCastNet, GraphCast, etc., where the models tend to learn the large-scale features better but struggle to accurately represent the small-scale fronts, filaments, atmospheric rivers, etc. Moreover, the high correlation coefficient in the instantaneous fluxes in Figure 7 show that most of these small values might be appearing outside of the selected hotspots. We propose a couple of solutions to tackle this issue:
> - A more physics-informed loss function, for e.g, regularizing the predicted distribution towards the true distribution
>
> - A latitude-weighted loss function – since the correlation coefficient is weaker in the tropics, a latitude weighted loss function can improve/reduce the errors in the tropics
>
> - A different scaling for fluxes - strong tendency for small values to be treated as noise. While the cuberoot scaling of fluxes works very well in yielding a plausible climatology, it may make it difficult to learn the small-scale values as the cuberoot tends to push small values away from zero and large-values towards one.
>
> 3. This is a good question, thanks! The traditional Hellinger distance ranges from 0 to 1, 0 indicating that the two distributions are identical for the whole sample space and 1 indicating that the two distributions are completely disjoint. The tail-Hellinger distance envisioned in this study, however, has a different range. We provided some context how to interpret values for the tail-Hellinger metric in the manuscript but would be happy to elaborate more in the revised version.
>
> - Essentially, tail-Hellinger zooms in on the tails assuming that the bulk of the distributions are similar (if not identical). With ½ as a fixed constant, a positive increment to tail-Hellinger come from the second term which is simply the integral of the tail for the predicted distribution. If the bulk is identical for both $p$ and $q$, then the second term (integral of $p$) should be equal to 0.5 as well. If subsequently the tails are disjoint, then the tail-Hellinger will be equal to 1. In this case, the interpretation will be that the bulk of the distributions are identical, yet the tails are totally different/disjoint.
>
>
> - Any negative contribution, that makes the tail-Hellinger negative comes from the cross-term between $p$ (predicted) and $q$ (truth). If the tails of the two distributions are disjoint then the contribution from the cross term will be zero and the tail-Hellinger will be nudged toward more positive values by the second term (scaled integral of $p$). On the other hand, a negative value would imply a fatter tail of the predicted distribution than the true distribution because $\sqrt{p(x)}\sqrt{q(x)} \geq \sqrt{p(x)}\sqrt{p(x)}$ over the tails and thus.
>
>
> - So, if the Hellinger distance is (say) -0.25, this means the model is underestimating the tails, i.e. $\sqrt{q(x)} > \sqrt{p(x)}$. If the bulks are identical, this would mean that the range of $q$’s tails is narrower than the range of $p$’s tails but since $q (x)$ > $p(x)$ $\forall x$, the cross-integral $\int \sqrt{q(x)}\sqrt{p(x)} dx$ > 0.5 + $\int \sqrt{p(x)}\sqrt{p(x)} dx$, leading to negative values. The more the prediction underestimated the tails, the more negative is the tail hellinger distance. The fatter the tail of $q$ relative to $p$, the more negative the tail-Hellinger. However, the tail could be fatter on either one side or both, and the tail-Hellinger would not be informative in suggesting which (limitation).
>
>
> - It is worth mentioning that, negative values of tail-Hellinger are more likely for small values of $\epsilon$ as that increases the probability of the bulk of the distributions to not be identical.
>
>
> - Ofcourse, this is the net sum of the tail product over both tails of the distributions - sort of like a two-sided student t-test. A more nuanced metric could be defined to focus on just the left or the right tail. Another possibility is to not assume similar bulks and assume systematic biases in the mean and account for such biases when computing the distributions distance. Since, it was not super relevant for this study, we did not discuss these ideas in the Appendix.

---

> > ### Comment · Reviewer_5F5c · 2024-11-24
> >
> > Thank you for your detailed responses. While I appreciate the thorough clarification of technical points and acknowledge the potential impact of this work for climate science, I maintain my position regarding the paper's fit for ICLR for several reasons:
> >
> > 1. Although ICLR has a track for physical science applications, submissions are still expected to advance machine learning methodology or demonstrate novel ML techniques. You mention ClimSim's recognition at NeurIPS 2023, but it's important to note that this was in the datasets track, where it was recognized for providing a novel and comprehensive dataset to foster progress in the field. It was not accepted as an original research article advancing ML methodology, which is what your current submission aims to be.
> >
> > 2. While the integration of foundation models with climate science is important work, the primary contributions appear to be: (a) Application of existing fine-tuning methods to a new domain; (b) Domain-specific evaluation metrics (tail-Hellinger distance, which is not an entirely novel metric, but its use here is novel); (c) Empirical results on gravity wave modeling
> > These contributions, while valuable, would be better suited for venues focused on climate modeling, where the domain impact can be properly evaluated by experts in climate science.
> >
> > 3. The clarifications provided in your rebuttal are very helpful but are not reflected in the current manuscript or appendix. A revised manuscript incorporating these points would be necessary to reconsider the evaluation.
> >
> > Given these considerations, I maintain my original score while acknowledging the potential impact of this work in its intended application domain.

---

### Note · Authors · 2025-02-22

**Comment:**

We are withdrawing this paper as this is submitted elsewhere now. Thanks for considering the submission.

Best,
Authors

**Withdrawal Confirmation:**

I have read and agree with the venue's withdrawal policy on behalf of myself and my co-authors.

---

### Meta-Review · Area_Chair_AHqc · 2024-12-17

**Metareview:**

The paper proposes using foundation models to better the climate models predictions and specifically focusing on modeling the gravity wave effects via fine tuning. Almost all the authors agree that the technical contribution from ML perspective is minimal. I agree with the assessment of the reviewers that ICLR is perhaps not the best venue for this work.

**Additional Comments On Reviewer Discussion:**

All the reviewers are inclined towards rejecting the paper, their primary concern being venue fit. I agree with the reviewers that the ML contribution of the work is minimal and that the paper will be received better at an alternate venue.

---

### Decision · Program_Chairs · 2025-01-22

Reject